# Mercury Exposure from the Consumption of Dietary Supplements Containing Vegetable, Cod Liver, and Shark Liver Oils

**DOI:** 10.3390/ijerph20032129

**Published:** 2023-01-24

**Authors:** Barbara Brodziak-Dopierała, Agnieszka Fischer, Martyna Chrzanowska, Bożena Ahnert

**Affiliations:** 1Department of Toxicology and Bioanalysis, Faculty of Pharmaceutical Science, Medical University of Silesia, 30 Ostrogórska Str., 41-200 Sosnowiec, Poland; 2Pharmacy, Apteka Blisko Ciebie, 37 Edukacji Str., 43-100 Tychy, Poland

**Keywords:** mercury, dietary supplements, vegetable oils, shark liver oil, cod liver oil, AAS

## Abstract

Vegetable and fish oils constitute a significant part of all dietary supplements. Due to increasing environmental pollution, the raw materials used for their production may be contaminated with toxic substances, including metals. The aim of the present study was to determine the mercury (Hg) content in vegetable oils, shark liver oils, and cod liver oils. The tests conducted were to help determine the level of mercury contamination of the tested preparations and the related potential threat to human health. The amount of Hg in the tested dietary supplements was compared, and the amount of the metal consumed at various times of use was determined. A total of 36 preparations of dietary supplements available on the Polish market were used for the study. The method of atomic absorption spectrometry using the amalgamation technique was used for the determinations (AMA 254, Altec, Czech Republic). Among the sample of all of the tested preparations, the Hg concentration ranged from 0.023 to 0.427 µg/kg, with an average of 0.165 µg/kg. Differences in Hg content in the various tested preparations (shark liver oil, cod liver oil, and vegetable oils) were statistically significant. The average concentration of Hg in the vegetable oils (0.218 µg/kg) was more than twice that of the cod liver oils (0.106 µg/kg) and shark liver oils (0.065 µg/kg). In none of the tested preparations did the amount of Hg exceed the acceptable standard for dietary supplements (0.10 mg/kg). The analysis showed that the Hg content in vegetable oils, shark liver oils, and fish oils from the Polish market is at a low level, guaranteeing the safety of their use, and as such, they do not pose a threat to health.

## 1. Introduction

Environmental pollution associated with industrial activity, exhaust emissions, and the use of chemicals affects the quality of raw materials of vegetable and fish origin. Contaminated ingredients used by manufacturers can lead to human exposure. The contamination of dietary supplements can also occur in the subsequent stages of their production [1,2].

The largest anthropogenic source of mercury (Hg) is the burning of coal and other fossil fuels. Each year, as a result of these processes, approximately 2000 tons of Hg are released into the environment [3,4]. Mercury can exist in three different chemical forms: elemental, inorganic, and organic. Methylmercury is an organic form of mercury. In humans, methylmercury is readily absorbed into the bloodstream and distributed throughout the body to sites where the brain is located, and in pregnant women, to the developing fetus [5].

Exposure to mercury can cause disturbances in the functioning of the body. Relatively low doses of Hg have an adverse effect on the central nervous system, impairing cognitive functions, such as the ability to learn, remember, and concentrate [6,7]. Numerous nervous disorders can also arise, such as speech disorders, deafness, paralysis of the muscles of the limbs, and even death [8,9,10,11]. Chronic exposure to Hg, by increasing the production of free radicals and reducing the activity of antioxidant enzymes (glutathione peroxidase, catalase, and superoxide dismutase), leads to increased oxidative stress and cardiovascular failure [12,13,14,15,16,17,18,19,20]. Hg exposure is associated with cancer risk although the data are contradictory. Mercury may promote carcinogenic risk through modulation of cell proliferation. Oxidative DNA damage, genotoxicity, and the epigenetic effects of Hg may be involved in cancer [21].

The growing health awareness of consumers and the desires to improve health and maintain well-being, as well as the easy accessibility and increasing demand for advertising, contribute to the high popularity of supplements [22,23]. According to statistical data, the percentage of people using dietary supplements is constantly increasing. For example, in Poland their use increased from almost 48 to 66.7% within 4 years (2016–2020). Currently, about 70% of Poles [24] and about 50% of the US population [25] report taking dietary supplements. Additionally, in Dubai, United Arab Emirates (UAE), the market for dietary supplements is growing every year. The number of health supplement products available in Dubai increased from 690 in 2014 to 800 in 2016 [26].

Dietary supplements compensate for deficits of substances that have a nutritional effect. Properly used, they have a health-promoting effect, but they can also have an adverse effect [27,28]. It has been suggested that the decision to introduce supplementation should not be the result of a trend, but rather should be preceded by a thorough analysis of nutrition and health [22]. Dietary supplements can interact with drugs and change their effects, for example, by interfering with absorption, excretion, or metabolism. They can also have a toxic effect on the body due to the impurities present in them [27,29]. Low levels of heavy metal contamination in conjunction with daily consumption result in cumulative levels of exposure [30].

The increase in demand means that the number and variety of supplementary preparations introduced to the market is growing. In the case of the Polish market, it is a relatively simple and quick process that is not subject to legal regulations that would guarantee the appropriate quality of the products [24].

Oils, both vegetable and fish, have health-promoting effects and can be used as dietary supplements. Vegetable oils are obtained from the fruits and seeds of various plant species. They consist of 95% triglycerides, with small amounts of mono- and diglycerides of higher fatty acids. The other ingredients are secondary, nontriglycerol compounds, such as phospholipids, sulfolipids, glycolipids, sphingolipids, waxes, polyphenols, phytosterols, hydrocarbons, vitamins, proteins, and pigments in the form of carotenoids and chlorophyll [31,32,33,34,35,36,37]. Among the oils of fish origin, shark liver oil and cod liver oil stand out. Cod liver oil is a liquid fat obtained from the fresh liver of fish from the cod family, mainly Atlantic cod (*Gadus morhua*). It is characterized by a high content of essential fatty acids (EFA) and the vitamins A and D. Shark liver oil is rich in alkylglycerols and squalene [38,39,40,41].

Mercury is an element that is commonly found in the environment as a result of anthropogenic activity. It is a toxic element, causing negative effects on human health. Legal regulations specify the permissible Hg content in various products, the use of which may lead to the intoxication of the body. *The Official Journal of the European Union* specifies the maximum residue levels for mercury compounds in certain products. According to the regulations, the level of Hg in oil seeds and fruits may not exceed 0.2 mg/kg, except for olives, where the level may not exceed 0.01 mg/kg [42]. In dietary supplements, the Hg content should not exceed 0.10 mg/kg [43].

The aim of the present study was to determine the Hg content in vegetable oils, shark liver oils, and cod liver oils. These products are popularly used on the Polish market as dietary supplements. They support, among other things, the functioning of the immune system, blood circulation, and brain; they protect and care for the skin; they lower blood cholesterol; and they have a beneficial effect on the liver [33,38,40]. The conducted tests determined whether the actual amount of mercury in the analyzed preparations exceeded the standards. On the basis of our analysis of the Hg concentrations in the tested samples, the potential health risks to consumers when the supplements are administered orally was estimated. The amounts of Hg in the tested preparations were compared, and the amount of metal consumed with the dietary supplements was determined for varying periods of their use.

## 2. Materials and Methods

The study included 36 dietary supplements containing vegetable oils (N = 18) and fish oils (N = 18), including shark liver (N = 6) and cod liver oils (N = 12). All of the tested vegetable oils were cold-pressed and unrefined. The preparations had different forms (capsule/liquid) and came from various producers popular on the Polish market. A detailed description of the tested supplements is presented in Table 1. All oils were purchased in Poland in pharmacies, drugstores, and herbal shops.

The total Hg content in the tested samples was determined by atomic absorption spectrometry (AAS) via the amalgamation technique, using an AMA 254 spectrometer (Altec, Praha, Czech Republic) [44,45]. The measurement conditions were as follow: a wavelength of 253.65 nm and an inlet pressure of 200–250 kPa, with oxygen as the carrier gas (O_2_ purity ≥ 99.5%). The duration of the individual stages of the analysis (in seconds) were: drying 200 s, decomposition 250 s, and measurement 90 s. The lower limit of detection (LOD) was 0.01 ng Hg [45].

In order to verify the correctness of the applied method, Hg determinations were taken from the reference material *Mixed Polish Herbs* (*INCT-MPH-2*, Institute of Nuclear Chemistry and Technology, Department of Analytical Chemistry, Warsaw, Poland). The mercury results for the 6 replicates were 0.0184 ± 0.0003 mg/kg, and the recovery was 102.3%. The AMA 254 analyzer does not require sample preparation before determination. Oil samples were collected using a DISCOVERY Comfort DV200 automatic, single-channel pipette with a 20–200 ul ejector (HTL LAB SOLUTIONS), which guaranteed the best repeatability. Three samples of 100 µL each were taken from each dietary supplement for the determinations. The recorded concentration of Hg for each preparation was the arithmetic mean of 3 measurement results. The obtained results were subjected to statistical analysis using Statistica 13.3 pl for the Windows operating system (Statsoft, Cracow, Poland). Due to the nonnormal distribution of samples (Shapiro–Wilk test), the discussion of the results is based on median values, and the following nonparametric tests were used to compare the statistical variability between the tested groups of samples: the Mann–Whitney U test (for 2 groups) and the Kruskal–Wallis (ANOVA on ranks) test (for 3 groups). The statistically significant level of probability was *p* ≤ 0.05.

## 3. Results

Table 2 presents the mercury content found in the samples of vegetable, shark liver, and cod liver oils. In all of the tested preparations, the Hg concentration ranged from 0.023 µg/kg to 0.427 µg/kg, with an average of 0.165 µg/kg. The value of the coefficient of variation (55.8%) indicated large variations between the samples in terms of Hg content. Differences in Hg content in all of the tested preparations (shark liver oil, cod liver oil, and vegetable oil) were statistically significant (*p* < 0.001).

A lower Hg content was found in the fish oils (0.088 µg/kg) compared to the vegetable oils (0.218 µg/kg); these differences were statistically significant (*p* < 0.001). The greatest dispersion of results around the mean value characterized the group of preparations containing vegetable oils. In these samples, the range of nonoutlier Hg concentrations reached a maximum of 0.427 µgHg/kg (Figure 1). The graphical distribution of Hg concentrations in cod liver oils, shark liver oils, and vegetable oils is presented in Figure 2. The average concentration of Hg in vegetable oils (0.218 µg/kg) was more than double that of cod liver oil (0.106 µg/kg) and shark liver oils (0.065 µg/kg).

Figure 3 shows the results of the assays of the Hg contents in the shark liver oil samples. The concentrations of Hg ranged from 0.023 µg/kg to 0.087 µg/kg. The highest Hg content was found in preparations No. 1 (0.087 µg/kg), No. 6 (0.077 µg/kg), and No. 2 (0.073 µg/kg). The lowest concentration of Hg was found in preparation No. 4 (0.023 µg/kg). 

A higher and more diversified Hg content was observed in the cod liver oils, as compared to the shark liver oils (see Figure 4). The highest concentration of Hg was found in preparation No. 7 (0.207 µg/kg) and No. 3 (0.205 µg/kg). The concentrations of Hg in preparations Nos. 1, 4, and 5 were slightly lower and amounted to about 0.180 µg/kg. The lowest concentration of Hg among the cod liver oils (0.030 µg/kg) was found in preparation No. 12 (see Figure 4). In the analyzed samples of vegetable oils (Figure 5), the highest concentration of Hg was found in cedar nut oil (0.427 µg/kg, preparation No. 14), followed by evening primrose oil (0.320 µg/kg, preparation No. 18). In the other tested samples of vegetable oils, the Hg contents were lower; they did not exceed 0.300 µg/kg. In the case of oil from the seeds of *Camelina sativa*, two different preparations (No. 5 and 10) were assessed. The Hg concentration in preparation No. 5 was 0.173 µg/kg, lower than that of preparation No. 10 (0.203 µg/kg). The lowest concentration of Hg in the group of vegetable oils was found in sample No. 16 (0.131 µg/kg), which was grape seed oil.

Based on the analysis of the test results, none of the tested dietary supplements exceeded the permissible level of mercury (0.10 mg/kg) [38].

In the next stage of the research, the hypothetical intake of Hg was analyzed. The daily dose of the preparation was determined according to the manufacturers’ instructions. For preparations in the form of capsules, depending on the preparation, it was an amount of 1–6 capsules. For liquid preparations, it was an amount of 5–25 mL (0.005–0.025 dm^3^). Considering the weight of a single capsule or the volume of a single dose of the liquid preparations, the daily, weekly, monthly, and annual mercury intakes were calculated (see Table 3). Among all of the analyzed preparations, the highest daily intake of Hg was recorded for a vegetable oil: organic soybean oil (preparation No. 7, 0.0045 µg), and the lowest (0.00001 µg) was recorded for cod liver oil (preparation No. 12). Among cod liver oils, the highest daily intake of Hg was characterized by preparation No. 5 (0.00076 µg), and the lowest by No. 12 (0.00001 µg).

Among the vegetable oils, the highest daily consumption of mercury was recorded for preparation No. 7(0.0045 µg, organic soybean oil), and the lowest for preparation No. 17 (0.0004 µg, sea buckthorn oil). The annual Hg intake for safflower oil (preparation No. 1) was 1.64 µg. An annual Hg intake of more than 1 µg/g was also found for soybean oil (preparation No. 7), wheat germ oil (preparation No. 9), and evening primrose oil (preparation No. 18). The annual intake of Hg in the case of the consumption of vegetable oils was many times higher than that in either the case of the shark liver oils or that of the cod liver oils. Among the dietary supplements containing shark liver oil, the highest daily intake of Hg was found for preparation No. 2 (0.000092 µg), and the lowest for preparation No. 5 (0.00002 µg). 

The Scientific Panel on Contaminants in the Food Chain (CONTAM) EFSA’s has set the tolerable weekly intake (TWI) for methylmercury at 1.3 µg/kg bw and for inorganic mercury at 4 µg/kg bw. [46]. For an adult with an average weight of 70 kg, the TWI for methylmercury is 91 µg, and that for inorganic mercury is 280 µg. None of the tested preparations exceeded the value of the tolerable weekly intake. The TWI values for the tested oil samples amounted to a maximum of 0.03% (for vegetable oils: safflower and soybean, preparations No. 1 and 7). In the case of the consumption of fish oils (cod liver oil and shark liver oil), the TWI values were lower than those of the vegetable oils (see Table 3).

## 4. Discussion 

The offerings of supplementary preparations available on the market is constantly increasing [22]. According to European data, this industry was valued at USD 14.95 billion in 2019, and profits were expected to increase by 2027. In Poland, in the years 2020–2024, an approximately 5% increase in the sales of dietary supplements is expected by each year [27]. Dietary supplements are generally considered safe by consumers. There is a belief that they can be taken without restrictions; therefore, they are often abused and taken by people without signs of deficiency [47]. Meanwhile, their unjustified and improper use may result in serious health consequences [18,25,29,31,48].

Vegetable oils are a valuable source of monounsaturated fatty acids, which help maintain the right proportions between “bad” and “good” cholesterol. They are used in the prevention of and therapy for skin inflammation, atherosclerosis, diabetes, and cancer. They help maintain the youthful appearance of the skin and the so-called vitality of the body [33,34,35,49].

Fish oils include shark liver and cod liver oil. The effect of cod liver oil is mainly based on its acids: eicosapentaenoic acid (EPA) and docosahexaenoic acid (DHA). Due to the effect essential fatty acids have on the development of the nervous system, its supplementation is particularly important in young children and pregnant women. In addition, cod liver oil has positive effects on the immune and circulatory systems, supports the proper functioning of the skeletal system and the eyes. Shark liver oil, on the other hand, mainly has antiviral and antibacterial properties, supports the functioning of the immune system, and reduces cholesterol levels [40,41,50,51].

In terms of safety, 11% of the dietary supplements available on the market in Poland were assessed in 2017–2020. It is advisable that the system of supervision over dietary supplements be made more rigorous and effective to ensure the safety of consumers’ health [47]. 

In dietary supplements, the maximum allowable level of mercury content must not exceed 0.10 mg/kg [42]. In none of the oil preparations that we tested did the mercury content exceed the standard value. 

In order to assess the safety on human health, the amount of mercury intake with the tested dietary supplements was estimated in the cases of daily, weekly, monthly, and annual use. The tested dietary supplements with the highest Hg content, taken in accordance with the recommended dose, provided only 0.03% TWI. Among all of the 36 tested preparations, none exceeded this value, which indicates that their consumption does not pose a threat to human health in terms of Hg content.

The number of literature reports on the mercury content of dietary supplements containing vegetable and animal oils is small. The results of Smutna et al. [50] indicate that the total mercury level in fish oils ranges from 0.013 to 2.03 μg/kg. In our work, this range was 0.023–0.207 μg/kg; there were significant differences, especially in the upper range of content. In studies conducted in New Zealand by Rucklidge and Shaw [51], measurements of mercury in preparations containing fish oil did not show values above the detection limit of the instrument (LOD = 10 μg/kg). On the other hand, the analysis of fish oil samples conducted in the USA by Floran et al. [52] showed mercury concentrations ranging from 6 to 12 μg/kg. The quoted values significantly exceed the results of the Hg content obtained in the study by Smutna et al. [50], as well as in the samples we assessed, including both fish oils and all dietary oil supplements. The Polish study by Krygier et al. [53] showed that the average Hg content in cold-pressed rapeseed oil was 0.82 μg/kg. Rapeseed oil was not a subject of our own research, while in all the analyzed samples of vegetable oils, both the average concentration of this metal and the maximum values were significantly lower than those found by Krygier et al. [53].

In addition, Zhu et al. [54] determined the content of various metals (nickel, cadmium, manganese, and lead) in edible vegetable oils commercially available in China. The lowest content of Mn was found in olive oil (0.113 µg/g), and the highest (0.556 µg/g) in peanut oil. In the case of Ni, peanut oil had the lowest content (0.026 µg/g) and sesame oil (0.075 µg/g) the highest. The highest Cd content was found in corn oil (8.43 ng/g), and the lowest (2.64 ng/g) was found in olive oil. Among the tested vegetable oils, olive oil contained the lowest concentrations of both Mn and Cd [54]. Additionally, in the olive oil we tested, the concentration of Hg (0.152 µg/kg) was lower than that found in other vegetable oils, and it was also lower than the average value for all of the tested vegetable oils combined (0.218 µg/kg), which may indicate that this type of oil, compared to others, contains lower amounts of heavy metals such as Mn, Cd, and Hg. Zhu et al. [54] also found that the content of cadmium in the edible oil samples was 0.152 µg/kg, which did not exceed the permissible limits. The same was true for Pb; the lowest level was in corn oil (0.009 µgPb/g), and the highest level was in sesame oil (0.018 µgPb/g). These studies determined that the consumption of vegetable oils, in the context of contamination with the analyzed metals, is safe. On the other hand, in the vegetable oils assessed by Karasakal [55] (walnut, sweet almond, soybean, bitter almond, and coconut oil), the content of Sb was 1.02–1.66 µg/g, and that of Sn was 23–35 µg/g, which indicates that the average daily intake for these metals exceeded the tolerable weekly intake, which may pose a health risk to consumers.

Ghane et al. [48] observed that, depending on the continent where the oils originate, the contents of toxic and physiological elements vary. Vegetable oils from the Americas and Africa had higher concentrations of Cd, As, and Pb, while higher concentrations of Fe, Zn, Cu, and Ni were found in the products from the Western Pacific and Eastern Mediterranean regions [54]. Most of the vegetable oils we assessed came from Poland. Some of them (e.g., cedar oil, rice oil, and olive oil) came from other European Union countries.

An analysis of the Hg content in dietary supplements containing both fish and vegetable oils was conducted by Augustsson et al. [23]. The Hg content in the fish oils was 1.1 µg/kg, and in supplements containing vegetable ingredients, it was twice as high (2.1 µgHg/kg) [25]. A similar trend, although lower levels Hg content, developed in our research. The concentrations of Hg in the vegetable oils that we tested (0.218µg/kg) was more than twice as high as those we found in the fish oils (0.088µg/kg). As in the studies on oils cited above [25,48,50,52,54,55], none of the samples of vegetable oils, cod liver oils, or shark liver oils that we assessed exceeded the statutory maximum concentration for mercury. Based on the research results, it can be concluded that the concentrations of Hg in the samples of dietary supplements do not exceed the permitted standards, and their consumption is not a significant source of exposure to Hg. Our research included selected samples of dietary supplements and in limited numbers, which may constitute a study limitation. Food products, especially those based on natural ingredients should be subject to a quality control process. Due to the changing levels of heavy metals in food products, it is advisable that research on the Hg content continues. 

## 5. Conclusions

The Hg contents in all of the tested dietary supplements was in the range of 0.023–0.427 μg/kg, and on average, 0.165 μg/kg. Statistically significant differences in Hg content were found between the vegetable oils, cod liver oils, and shark liver oils. The Hg content in the fish oils was twice as low as that in the supplements containing vegetable ingredients.

In the tested dietary supplements, the highest amount of Hg was recorded in vegetable oils. The highest mercury content was found in cedar nut oil, and the lowest was found in grape seed oil. The smallest amount of Hg among all of the oil dietary supplements was found in shark liver oil. 

The determined Hg content in all of the tested dietary supplements did not exceed the maximum levels of contaminants in foodstuffs.

Based on the calculated values of the hypothetical mercury intake in the case of the daily, weekly, monthly, and annual use of the tested preparations, it was shown that the amount of mercury supplied with the tested dietary supplements available on the Polish market neither exceeds the TWI value nor poses a threat to the health of the consumers.

## Figures and Tables

**Figure 1 ijerph-20-02129-f001:**
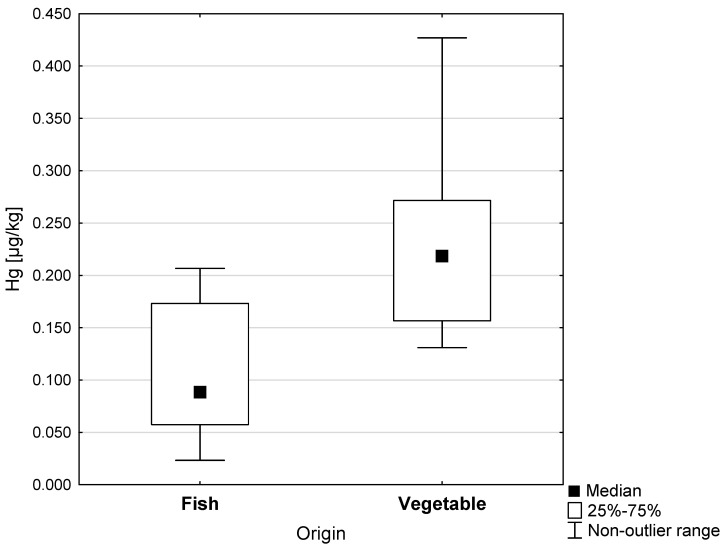
Hg concentrations in dietary supplements made of fish and vegetable oils (µg/kg).

**Figure 2 ijerph-20-02129-f002:**
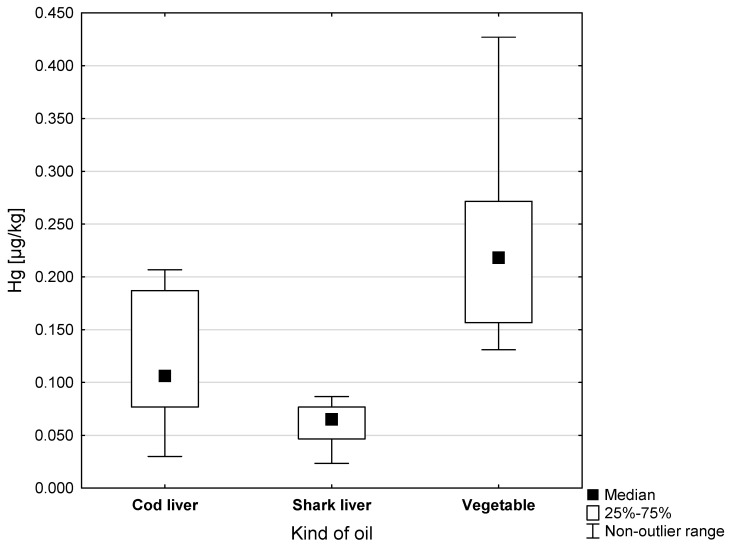
Hg concentrations in vegetable oils, cod liver oils, and shark liver oils (µg/kg).

**Figure 3 ijerph-20-02129-f003:**
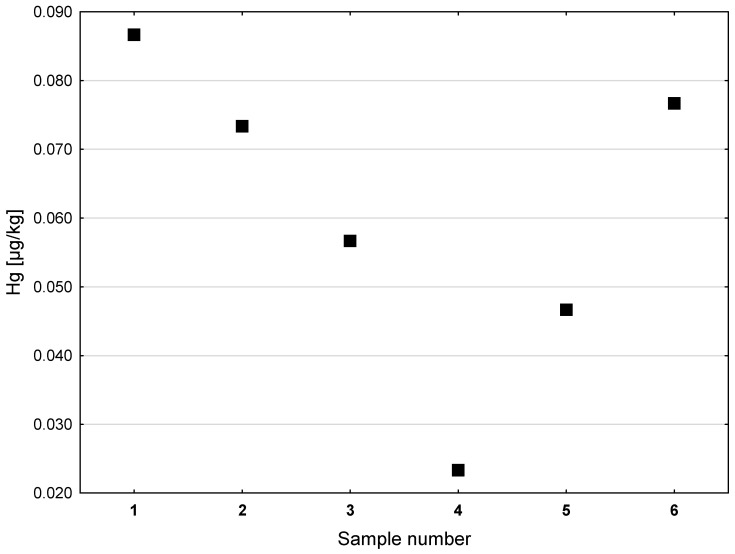
Hg concentrations in dietary supplements made with shark liver oil (µg/kg).

**Figure 4 ijerph-20-02129-f004:**
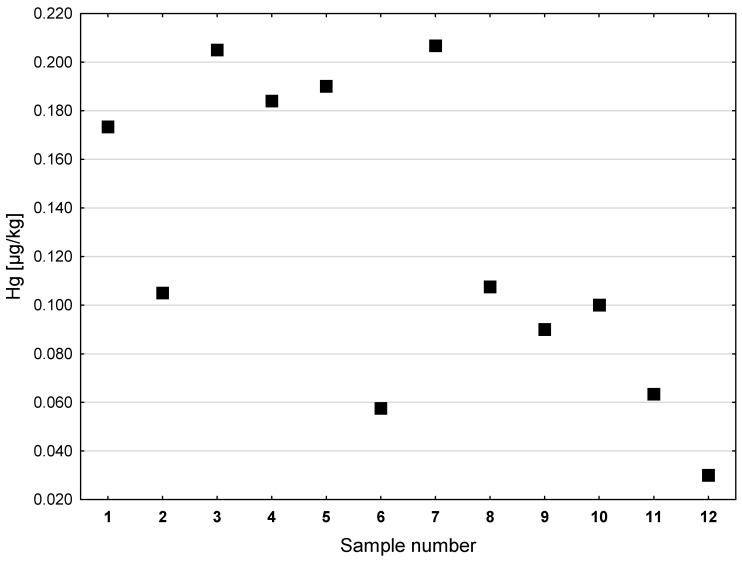
Hg concentrations in dietary supplements made with cod liver oil (µg/kg).

**Figure 5 ijerph-20-02129-f005:**
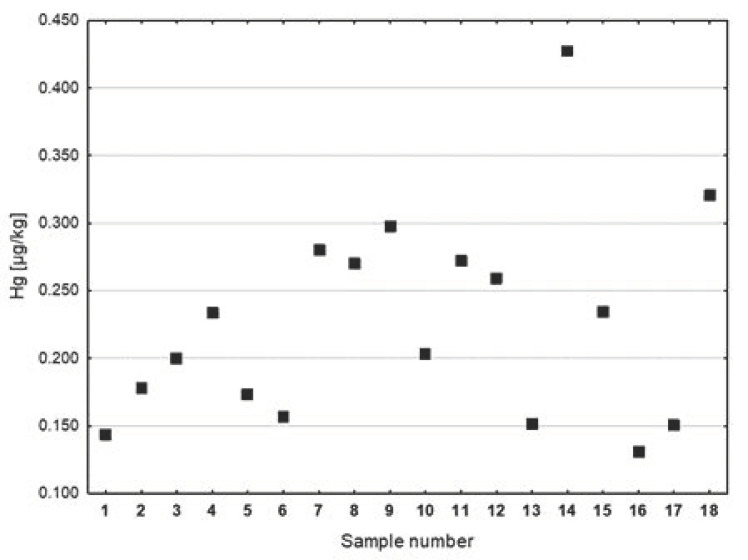
Hg concentrations in dietary supplements made with vegetable oil (µg/kg).

**Table 1 ijerph-20-02129-t001:** Characteristics of the tested dietary supplements.

Preparation		Form	Main Component	Composition/Description
Number	Name			
COD LIVER OILS (N = 12)				
1	Mega Tran	capsule	cod liver oil	DHA acid, EPA acid, vitamins A, D
2	Doppelherz Aktiv Tran	capsule	cod liver oil	DHA acid, EPA acid, vitamins A, D
3	Tran	capsule	cod liver oil	DHA acid, EPA acid, vitamins A, D
4	Moller’s Forte	capsule	cod liver oil	DHA acid, EPA acid, vitamins A, D
5	Moller’s Tran Norweski	liquid	cod liver oil, vit. E	DHA acid, EPA acid, vitamins A, D
6	Tran Islandzki LYSI	liquid	cod liver oil, vit. E	DHA acid, EPA acid, vitamins A, D
7	Tran Vitamex	capsule	cod liver oil	DHA acid, EPA acid, vitamins A, D
8	Tran Hasco	capsule	cod liver oil	DHA acid, EPA acid, vitamins A, D
9	Omega 3-6-9	capsule	cod liver oil, linseed oil, borage	DHA acid, EPA acid
10	Vitter Blue Tran	capsule	cod liver oil	DHA acid, EPA acid, vitamins A, D
11	Tran z Witaminami	capsule	cod liver oil, vit. E	DHA acid, EPA acid, vitamins A, D
12	Life Tran z Wątroby Dorsza	capsule	cod liver oil	DHA acid, EPA acid, vitamins A, D
SHARK LIVER OILS (N = 6)				
1	Iskial	capsule	shark liver oil, vitamin D	alkylglycerols, squalene
2	Rekinol	capsule	shark liver oil, garlic extract, vitamin D	alkylglycerols
3	Ecomer	capsule	shark liver oil, vitamin E	alkylglycerols
4	Life Shark Liver Oil	capsule	shark liver oil	alkylglycerols
5	Max Tran Plus	capsule	shark liver oil, vitamins A, D, E	alkylglycerols, squalene
6	Iskial Plus	capsule	shark liver oil, vitamin D, garlic extract	alkylglycerols, squalene
VEGETABLE OILS (N = 18)				
1	Safflower Oil	liquid	safflower oil	cold-pressed, unrefined
2	Corn Oil	liquid	corn germ oil	cold-pressed, unrefined
3	Linseed Oil	liquid	linseed oil	cold-pressed, unrefined
4	Poppyseed Oil	liquid	poppyseed oil	cold-pressed, unrefined
5	*Camelina Sativa* Oil	liquid	*Camelina sativa* seed oil	cold-pressed, unrefined
6	Rice Oil	liquid	rice oil	cold-pressed, unrefined
7	Organic Soybean Oil	liquid	soybean seed oil	cold-pressed, unrefined
8	Amaranth Oil	liquid	amaranth seed oil	cold-pressed, unrefined
9	Wheat Germ Oil	liquid	wheat germ oil	cold-pressed, unrefined
10	*Camelina Sativa* Oil	liquid	*Camelina sativa* seed oil	cold-pressed, unrefined
11	*Nigella Sativa* Seed Oil	liquid	nigella seed oil	cold-pressed, unrefined
12	Organic Nettle Seed Oil	liquid	nettle seed oil	cold-pressed, unrefined
13	Olive Oil	liquid	olive oil	cold-pressed, unrefined
14	Cedar Nut Oil	liquid	cedar nut oil, vitamin E	cold-pressed, unrefined
15	Thistle Oil	liquid	milk thistle oil	cold-pressed, unrefined
16	Grape Seed Oil	liquid	grape seed oil	cold-pressed, unrefined
17	Sea Buckthorn Oil	liquid	sea buckthorn oil	cold-pressed, unrefined
18	Evening Primrose Oil	liquid	evening primrose seed oil	cold-pressed, unrefined

**Table 2 ijerph-20-02129-t002:** Hg concentrations in the tested dietary supplements (µg/kg).

Dietary Supplement		N	Arithmetic Mean ± SD	Minimum	Maximum	Median	Quartile		Coefficient of Variation (%)	*p*-Level
							Q1	Q3		
All		36	0.165 ± 0.092	0.023	0.427	0.165	0.088	0.220	55.8	
Type of oil	Cod liver	12	0.126 ± 0.063	0.030	0.207	0.106	0.077	0.187	49.6	<0.001
	Shark liver	6	0.061 ± 0.023	0.023	0.087	0.065	0.047	0.077	38.4	
	Vegetable	18	0.227 ± 0.077	0.131	0.427	0.218	0.157	0.272	33.9	

N—the number of samples; SD—standard deviation; *p*-level estimated using the ANOVA on ranks (Kruskal–Wallis) test.

**Table 3 ijerph-20-02129-t003:** Mercury supply from dietary supplements in the case of daily, weekly, monthly, and annual use.

Preparation		Recommended Daily Dose	Hg		Intake [µg]				% TWI
Number	Name		In Sample (µg/g)	In Single Dose (µg)	Daily	Weekly	Monthly	Annually	
COD LIVER OILS (N = 12)									
1	Mega Tran	2 capsules (540 mg)	0.00017	0.00005	0.00009	0.00064	0.0028	0.034	0.0007
2	Doppelherz Aktiv	4 capsules (1200 mg)	0.00011	0.00003	0.00013	0.00092	0.0040	0.048	0.0010
3	Tran	2 capsules (500 mg)	0.00021	0.00005	0.00011	0.00074	0.0032	0.039	0.0008
4	Moller’s FORTE z Tranem	2 capsules (600 mg)	0.00018	0.00005	0.00011	0.00076	0.0032	0.039	0.0008
5	Moller’s Tran Norweski	5 mL	0.00019	0.00076	0.00076	0.00532	0.0228	0.277	0.0060
6	Tran Islandzki LYSI	5 mL	0.00006	0.00024	0.00024	0.00168	0.0072	0.088	0.0020
7	Tran w Kapsułkach Vitamex	2 capsules (800 mg)	0.00021	0.00008	0.00017	0.00118	0.0050	0.061	0.0010
8	Tran Hasco	2 capsules (1000 mg)	0.00011	0.00006	0.00011	0.00077	0.0033	0.040	0.0008
9	Omega 3-6-9	1 capsule (500 mg)	0.00009	0.00005	0.00005	0.00032	0.0014	0.016	0.0003
10	Vitter Blue Tran	1 capsule (700 mg)	0.00010	0.00007	0.00007	0.00049	0.0021	0.026	0.0005
11	Tran z Witaminami	1 capsule (500 mg)	0.00006	0.00003	0.00003	0.00021	0.0009	0.011	0.0002
12	Life Cod-Liver Oil	1 capsule (400 mg)	0.00003	0.00001	0.00001	0.00008	0.0004	0.004	0.0001
SHARK LIVER OILS (N = 6)									
1	Iskial Plus	4 capsules (960 mg)	0.00008	0.000019	0.000076	0.00053	0.0023	0.028	0.0005
2	Iskial	4 capsules	0.00009	0.000023	0.000092	0.00064	0.0028	0.034	0.0007
3	Rekinol	2 capsules (1000 mg)	0.00007	0.000035	0.000070	0.00049	0.0021	0.026	0.0005
4	Ecomer	6 capsules (1500 mg)	0.00006	0.000015	0.000090	0.00063	0.0027	0.033	0.0007
5	Life Olej z Wątroby Rekina	2 capsules (1000 mg)	0.00002	0.000010	0.000020	0.00014	0.0006	0.007	0.0001
6	Max Tran Plus	2 capsules (500 mg)	0.00005	0.000013	0.000026	0.00018	0.0008	0.009	0.0002
VEGETABLE OILS (N = 18)									
1	Safflower Oil	20 mL	0.00028	0.0045	0.0045	0.032	0.135	1.64	0.030
2	Corn Oil	5 mL	0.00027	0.0011	0.0011	0.008	0.033	0.40	0.008
3	Linseed Oil	10 mL	0.00026	0.0021	0.0021	0.015	0.063	0.77	0.020
4	Poppyseed Oil	5 mL	0.00027	0.0011	0.0011	0.008	0.033	0.40	0.008
5	*Camelina Sativa* Oil	15 mL	0.00014	0.0017	0.0017	0.012	0.051	0.62	0.010
6	Rice Oil	15 mL	0.00018	0.0022	0.0022	0.015	0.066	0.80	0.020
7	Organic Soybean Oil	25 mL	0.00020	0.0045	0.0045	0.028	0.120	1.46	0.030
8	Amaranth Oil	10 mL	0.00020	0.0016	0.0016	0.011	0.048	0.58	0.010
9	Wheat Germ Oil	15 mL	0.00023	0.0028	0.0028	0.020	0.084	1.02	0.020
10	*Camelina Sativa* Oil	15 mL	0.00015	0.0018	0.0018	0.013	0.054	0.66	0.010
11	*Nigella Sativa* Seed Oil	15 mL	0.00013	0.0016	0.0016	0.011	0.048	0.58	0.002
12	Organic Nettle Seed Oil	10 mL	0.00016	0.0013	0.0013	0.009	0.039	0.47	0.010
13	Olive Oil	6 mL	0.00015	0.0007	0.0007	0.005	0.021	0.26	0.005
14	Cedar Nut Oil	5 mL	0.00043	0.0017	0.0017	0.012	0.051	0.62	0.010
15	Thistle Oil	5 mL	0.00032	0.0013	0.0013	0.009	0.039	0.47	0.010
16	Grape Seed Oil	5 mL	0.00030	0.0012	0.0012	0.008	0.036	0.44	0.009
17	Sea Buckthorn Oil	5 mL	0.00010	0.0004	0.0004	0.003	0.012	0.15	0.003
18	Evening Primrose Oil	15 mL	0.00023	0.0028	0.0028	0.020	0.084	1.02	0.022

N—the number of samples; TWI—tolerable weekly intake.

## Data Availability

Not applicable.

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
