# Peer review of "Mercury Exposure from the Consumption of Dietary Supplements Containing Vegetable, Cod Liver, and Shark Liver Oils"

_ijerph, 2023, doi:10.3390/ijerph20032129_

Round 1

Reviewer 1 Report

1)      Line 167-177: This part should be added in the materials and methods part instead of the results.

2)      Line 200-201: According to European data, this industry was valued at USD 14.95 billion 200 in 2019 and profits are expected to increase by 2027.

 Please provide the reference and citation

3)      Line 222-223: In Poland, the assessment of the safety of dietary supplement preparations is carried 222 out by the State Sanitary Inspection. In 2017-2020, out of approx. 63 000 preparations, 223 tested only 11%.

Need to rephrase the sentence 

4)      Line 258-281: It seems irrelevant according to the data presented. Therefore, it must be discussed by correlating it with mercury.

5)      Discussion part may be approved by adding new information.

6)      Conclusion should be concise and should be conclusively explained in one paragraph. 

Author Response

I would like to thank the Reviewers for the assessment of our manuscript, submitted comments
and indicated errors

In response to the review I would like to inform:

Reviewer #1:

1) Line 167-177: This part be added in the materials and methods.

2) Line 200-201: The sentence was changed.

3) Line 222-223: The sentence was changed and reference was added.

4) Line 258-281: Information on other metals has been removed.

5) The Discussion section was changed and supplemented.

6) The Conclusions section were changed and explained in one paragraph.

We hope that the explanations and changes which were made will prove satisfactory for the Reviewers and will allow accepting the manuscript for publication.

Best regards,

Reviewer 2 Report

see the pdf file with comments

Author Response

I would like to thank the Reviewers for the assessment of our manuscript, submitted comments and indicated errors

In response to the review I would like to inform:

Reviewer #2:

The changes and comments have been made in the pdf file.

We hope that the explanations and changes which were made will prove satisfactory for the Reviewers and will allow accepting the manuscript for publication.

Best regards,

Authors of the manuscript

Reviewer 3 Report

'The conducted tests determined whether the actual amount of mercury in the analyzed preparations did not exceed the permissible standards.'

[Comment: Where are your comparison to the maximum permissible limit of total Hg? Are there any guidelines in the edible oils? Those for shellfish are well established.

Suggest to include: THQ and carcinogenic risk of Hg for the samples.

Are the total elemental Hg related to methyl-Hg? Why? Methyl-Hg is the most toxic one.

Good data with high data validation of CRM recovery.

Suggest to compare more updated or outdated data of the Hg in the edible oil samples.

Tolerable Weekly Intake (TWI) or PTWI for Hg is valid but suggest to discuss more on this human health assessment.

Author Response

I would like to thank the Reviewers for the assessment of our manuscript, submitted comments and indicated errors

In response to the review I would like to inform:

Reviewer #3:

As regards mercury EFSA adopted on 24 February 2004 an opinion related to mercury and methylmercury in food and endorsed the provisional tolerable weekly intake of 1,6 μg/kg bw. Methylmercury is the chemical form of most concern and can make up more than 90 % of the total mercury in fish and seafood. The forms of mercury present in these other foods (than fish and seafood) are mainly not methylmercury and they are therefore considered to be of lower risk [42].On the basis of multiple epidemiological studies EFSA revised its former opinion on methyl mercury proposing a tolerable weekly intake (TWI) of 1.3 μg kg-1 bw week-1 lower than the former one (EFSA 2012) [46].[42] Commission Regulation (EU) 629/2008 of 2 July 2008 r. amending Regulation (EC) No 1881/2006 setting maximum levels for certain contaminants in foodstuffs (accessed on 5 September 2022).

[46] Mercury in food - EFSA updates advice on risk for public health. URL: https://www.efsa.europa.eu/en/press/news/121220 (accesed on 15 September 2022).

 Study products (oils, both vegetable and fish) have a health-promoting effect and are popularly used by Polish people. Thank you for your suggestion to include THQ, which we will be very happy to use in future research. 

The risk of mercury carcinogenicity has been indicated in the content of the manuscript.

Hg exposure is associated with cancer risk, although the data are contradictory. Mercury may promote carcinogenic risk through modulation of cell proliferation. Oxidative DNA damage, genotoxicity, epigenetic effects of Hg may be involved in cancer [21].

[21] Skalny A.V., Aschner M., Sekacheva M.I., Santamaria A., Barbosa F., Ferrer B., Aaseth J., Paoliello M.MB., Rocha J.B.T., Tinkov A.A. Mercury and cancer: Where are we now after two decades of research? Food and Chemical Toxicology 2022, DOI: https://doi.org/10.1016/j.fct.2022.113001

Mercury can exist in three different chemical forms: elemental, inorganic and organic). Methylmercury is an organic form of mercury and is very toxic at high levels of exposure. It’s the most common form of mercury in fish, is present in some types of fish at concentrations that could potentially impair human health. In humans, methylmercury is readily absorbed into the bloodstream and distributed throughout the body to sites where the brain is located, and in pregnant women, to the developing fetus [5].

[5] https://www.canada.ca/en/health-canada/services/food-nutrition/food-safety/chemical-contaminants/environmental-contaminants/mercury/mercury-fish-questions-answers.html#s1 The literature reports on the mercury content in edible oils using as dietary supplements are small.Our research included selected samples of dietary supplements and their limited numbers, which may constitute a study limitation. Food products, especially those based on natural ingredients should be subject to the control process.

We agree the human health assessment is very important. Due to the changing level of heavy metals in food products it is advisable to continue research on the content of Hg.

We hope that the explanations and changes which were made will prove satisfactory for the Reviewers and will allow accepting the manuscript for publication.

Best regards,

Authors of the manuscript

Reviewer 4 Report

General comment:

The manuscript entitled “Mercury exposure from consumption of dietary supplements containing vegetable oils, cod liver oil and shark liver oil” brings useful information about the mercury (Hg) content in vegetable oils, shark liver oils and cod liver oils collected from the Polish market and determined the potential threat to human health. The authors claimed that in all tested samples, the amount of Hg did not exceed the acceptable standard for dietary supplements and therefore they ensured the safety of the use of dietary supplements containing vegetable oils, cod liver oil and shark liver oil available in the Polish market. The reviewer believed that the present study is interesting and potentially could contribute to the research field, however, there are some concerns and questions that require be addressed to clarity and improve the present version.

Specific comments:

1. In the abstract, authors should follow the rules of abbreviation. For example, first-time mercury (Hg) should be written, then from next time, only Hg should be written.

2. The introduction largely failed to provide sufficient background information. There are several studies that investigated more than one heavy metal in dietary supplement products using a large number of samples (i.e., https://doi.org/10.1038/s41598-020-76000-w ; https://doi.org/10.1186/s12906-019-2693-3). Why did the authors of the current study only choose Hg for their study using a relatively small number of samples?

3. In line 34, the statement “The largest anthropogenic source of mercury (Hg) is the burning of coal and other fossil fuels” needs appropriate citations.

4. In the materials and methods section, detailed information on sample preparation as well as QA and QC should be added.

5. In the results section, the results are not well-organized and also not well-explained. Please compare your results with other previously published reports.

6. Discussion should be more focused, and the authors should discuss their findings from multiple angles. Please add a paragraph on the limitations of this study.

7. In conclusion, opportunities to inform future research were not properly addressed.  

Author Response

I would like to thank the Reviewers for the assessment of our manuscript, submitted comments
and indicated errors

In response to the review I would like to inform:

Reviewer #4:

  1. In the abstract, the abbreviation Hg was used.
  2. 2. The small number of samples resulted from the availability of animal oil preparations on the Polish market and the specificity of mercury determinations in these samples. The majority of dietary supplements in Poland are plant or mineral preparations. Mercury determinations are only caused by the possibilities of the apparatus (AMA 254 spectrometer Altec, Praha, Czech Republic).

The listed works have been cited:

[26] Abdulla, N.M.; Adam, B.; Blair, I.; Oulhaj, A. Heavy metal content of herbal health supplement products in Dubai – UAE: a cross-sectional study. BMC Complementary and Alternative Med. 2019. doi.org/10.1186/s12906-019-2693-3

[30] Jairoun, A.A.; Shahwan, M.; Zyoud, S.H. Heavy metal contamination of dietary supplements products available in the UAE markets and the associated risk. Scientific Reports. 2020, 10: 18824. https://doi.org/10.1038/s41598-020-76000-w

  1. In line 34, reference has been added
  2. The analyzer uses different control levels. The internal system provides QC/QA quality procedures.
  3. The our results were compare with other previously published reports in the Discussion section.
  4. The Discussion section was changed and supplemented.

The limitations of this study was added.

Our research included selected samples of dietary supplements and their limited numbers, which may constitute a study limitation. Food products, especially those based on natural ingredients should be subject to the control process. Due to the changing level of heavy metals in food products it is advisable to continue research on the content of Hg.

  1. The Conclusion section was changed.

We hope that the explanations and changes which were made will prove satisfactory for the Reviewers and will allow accepting the manuscript for publication.

Best regards,

Authors of the manuscript

Round 2

Reviewer 4 Report

Reviewer's comments were partially addressed. A round of a minor revision is suggested.  

For revision, please follow previous comments ---

-In the materials and methods section, detailed information on sample preparation should be added.

-In the results section, the results are not well-organized and also not well-explained. Please compare your results with other previously published reports.

Author Response

  1. The AMA 254 analyser does not require sample preparation before determination. Oil samples were collected using an automatic single-channel pipette DISCOVERY Comfort DV200 with a 20-200ul ejector (HTL LAB SOLUTIONS), which guarantees the highest repeatability.

  1. I find it difficult to refer to the Reviewer's comments regarding compare results with other published reports in the Result section.

The instructions for authors do not contain such an indication (Results: Provide a concise and precise description of the experimental results, their interpretation as well as the experimental conclusions that can be drawn). According to the requirements of the journal: https://www.mdpi.com/ journal/ijerph/instructions#preparation

The results of your own research with others should be posted in the Discussion section. (Discussion: Authors should discuss the results and how they can be interpreted in perspective of previous studies and of the working hypotheses. The findings and their implications should be discussed in the broadest context possible and limitations of the work highlighted. Future research directions may also be mentioned. This section may be combined with Results.)

Please forgive me if my explanations prove insufficient
